# The Projective Consciousness Model: Projective Geometry at the Core of Consciousness and the Integration of Perception, Imagination, Motivation, Emotion, Social Cognition and Action

**DOI:** 10.3390/brainsci13101435

**Published:** 2023-10-09

**Authors:** David Rudrauf, Grégoire Sergeant-Perthuis, Yvain Tisserand, Germain Poloudenny, Kenneth Williford, Michel-Ange Amorim

**Affiliations:** 1CIAMS, Université Paris-Saclay, 91405 Orsay, France; michel-ange.amorim@universite-paris-saclay.fr; 2CIAMS, Université d’Orléans, 45067 Orléans, France; 3Laboratoire de Biologie Computationnelle et Quantitative (LCQB), CNRS, IBPS, UMR 7238, Sorbonne Université, 75005 Paris, France; gregoireserper@gmail.com; 4IMJ-PRG, Inria Paris-Ouragan Project-Team, Sorbonne University, 75005 Paris, France; 5CISA, Université de Genève, 1202 Genève, Switzerland; yvain.tisserand@unige.ch; 6Laboratoire de Mathématiques de Lens (LML), UR 2462, Université d’Artois, 62300 Lens, France; germain_poloudenny@ens.univ-artois.fr; 7Philosophy and Humanities, University of Texas at Arlington, Arlington, TX 76019, USA; williford@uta.edu

**Keywords:** consciousness, computational modeling, projective geometry, active inference, affective value, epistemic value, emotion, social cognition and communication, behavioral science

## Abstract

Consciousness has been described as acting as a global workspace that integrates perception, imagination, emotion and action programming for adaptive decision making. The mechanisms of this workspace and their relationships to the phenomenology of consciousness need to be further specified. Much research in this area has focused on the neural correlates of consciousness, but, arguably, computational modeling can better be used toward this aim. According to the Projective Consciousness Model (PCM), consciousness is structured as a viewpoint-organized, internal space, relying on 3D projective geometry and governed by the action of the Projective Group as part of a process of active inference. The geometry induces a group-structured subjective perspective on an encoded world model, enabling adaptive perspective taking in agents. Here, we review and discuss the PCM. We emphasize the role of projective mechanisms in perception and the appraisal of affective and epistemic values as tied to the motivation of action, under an optimization process of Free Energy minimization, or more generally stochastic optimal control. We discuss how these mechanisms enable us to model and simulate group-structured drives in the context of social cognition and to understand the mechanisms underpinning empathy, emotion expression and regulation, and approach–avoidance behaviors. We review previous results, drawing on applications in robotics and virtual humans. We briefly discuss future axes of research relating to applications of the model to simulation- and model-based behavioral science, geometrically structured artificial neural networks, the relevance of the approach for explainable AI and human–machine interactions, and the study of the neural correlates of consciousness.

## 1. Introduction

The pursuit of a well-motivated, operational, and falsifiable theory of consciousness remains hot in cognitive science (see [1]). Such a theory holds the key to answering fundamental questions in psychology, neuroscience, cybernetics, artificial intelligence and robotics. A wealth of proposals of varying degrees of precision and heuristic value have flourished over the years. Yet there remains no consensus about which contender might be most promising. With the development of cognitive neuroscience and related empirical approaches (electrophysiology, neuroimaging), a great deal of research has been focused on the neural correlates of consciousness (NCC) as the presumed shining path to understanding its underlying mechanisms [2,3,4]. To be sure, understanding the relationships between consciousness and the brain would not merely improve our understanding of consciousness itself and its relations to behavior; it also seems to be a necessary condition for a complete theory of consciousness considered as part of nature. However, there is no reason to restrict research on consciousness to the study of its neural mechanisms. Investigating the principles and mechanisms that constitute consciousness irrespective of their neural implementation could set the stage for more theoretically grounded and model-based research in cognitive neuroscience, with well-posed quantitative hypotheses [5], while mitigating methodological roadblocks besetting the search for the NCC and neurally grounded models of consciousness more generally [6]. From this perspective, purely phenomenological and functional postulates offer relevant starting points for the formulation of a mathematical theory of consciousness.

### 1.1. Consciousness as an Integrative Whole

One pervasive intuition about the phenomenology and function of consciousness is that it integrates a multiplicity of cognitive functions and mechanisms into a coherent whole in order to facilitate cognition, learning, and adaptive behaviors or, more generally, to perform a cybernetic function for adaptive systems [7]. It integrates and mediates the interplay of processes such as perception, imagination, emotion, affective and epistemic (curiosity-related) drives, social cognition and action planning to leverage both exploration and exploitation behaviors.

Two recent, prominent theories embrace such an integrative view. Integrated Information Theory (IIT) conceives of the relevant whole as corresponding to the dynamical ensemble (or complex) of interactions between a system’s parts (e.g., brain networks) that maximizes the quantity of information in such ensembles, measured in terms of Φ, which cannot be reduced to the sum of information contained in those parts when considered independently [8]. In other words, IIT operationalizes the notion that the whole is more than the sum of its parts. IIT remains quite general; it predicts that some very simple physical systems (e.g., two suitably connected photo-diodes) are conscious [5,9]. Moreover, its formalism is difficult to apply in significant simulations and falsifiable empirical research.

The Global Workspace Theory (GWT) [10,11] conceives of consciousness as an integrative workspace featuring limited capacity and serial processing for decision making. The workspace accesses and broadcasts salient multimodal sensory information and combines it with information from memory. It supports the monitoring and reduction of uncertainty and error-correction mechanisms. It performs non-social and social imaginary simulations and appraises their outcomes. Its core function is to support planning, decision-making, and action programming. GWT has been modeled using “toy” models of neural networks in an analogical manner in conjunction with empirical research using brain imaging [12,13,14]. However, GWT has not offered mathematical principles or models capable of capturing the ensemble of functions it considers for consciousness, let alone the mechanisms of their interaction. Attempts at mathematical modeling of GWT, though quite interesting, have remained rather generic, have been based on information-theoretic concepts, and have focused on neurally relevant notions [15]. However, they have not integrated the type of geometrical perspective we see as being central to consciousness; and they cannot be straightforwardly operationalized to generate simulations relating the GW, cognitive and affective processing, and behavior.

Furthermore, formal expressions of IIT and GWT have not been derived in a way sufficiently specific to enable the direct comparison of their predictions; instead indirect and rather non-specific hypotheses have been proposed to assess their relative worth, focusing on whether the NCC involves anterior versus only posterior regions of the brain [16]. The debate is far from settled (see [9,17,18]). Similar limitations related to a lack of specificity and discrepancies between levels of observation arise when considering other theoretical proposals, e.g., using the General Theory of Information (GTI) [19,20].

### 1.2. The Subjective Perspective of Consciousness

Another pervasive intuition about the phenomenology and function of consciousness is the constitutive role played at its core by a “subjective perspective”. Such an internal perspective has been conceived of as a non-trivial, viewpoint-organized, unified, embodied three-dimensional (3D) representation of the world in perspective [8,21,22,23,24,25].

This subjective perspective is often referred to as a first-person perspective (1PP) in the literature on visuo-spatial perspective taking [26,27,28]. Moreover, as we will understand the matter here, seeing the world through someone else’s eyes or imagining ourselves from an external observer perspective corresponds to adopting a third-person perspective (3PP) [26,27,28]. Although a few authors also use egocentric and “altercentric” perspectives to refer to 1PP and 3PP, respectively [29,30], we sometimes use the latter abbreviations in this article. Note that whichever subjective perspective is adopted, its content is always a subjective perspective, somehow echoing Merleau-Ponty [31], writing: “I am a consciousness, a strange creature which resides nowhere and can be everywhere present in intention” (p. 43), and “[…] if the spatio-temporal horizons could, even theoretically, be made explicit and the world conceived from no point of view, then nothing would exist; I should hover above the world, so that all times and places, far from becoming simultaneously real, would become unreal, because I should live in none of them and would be involved nowhere. If I am at all times and everywhere, then I am at no time and nowhere” (p. 387). Thus, the concepts of 1PP and 3PP as we use them here do not correspond to their frequent use, sometimes including also a second-person perspective (2PP), in consciousness studies, e.g., in neurophenomenology, to distinguish consciousness as experienced directly from a 1PP, from consciousness as it can be studied indirectly from a 3PP, for instance, to study its NCC.

One of the key functional roles of this subjective perspective would be to enable situated systems [32] imbued with consciousness to take different perspectives, through imagination or action, in order to appraise affordances and maximize utilities at multiple time scales [25,33,34]. The process would combine (spatial, interoceptive and exteroceptive) cognitive and affective representations for action programming [35,36]. Perspective taking would also support social cognition, including empathy and Theory of Mind (ToM), which rely on the ability to infer the mental states of others, especially their beliefs and desires, and to predict their behaviors [36,37,38,39]. Consistent with our approach, simulation theory hypothesizes that humans use their own cognitive and affective functions to imagine themselves in the “shoes” of others, to simulate their subjective experience and infer the corresponding expected behaviors [40,41,42,43,44]. Modeling such subjective perspectives is an essential challenge for consciousness science [45,46,47,48,49,50].

Some, including GWT proponents, have decided to set the challenge aside [2,11], while others, in particular IIT proponents, have attempted to tackle it based purely on information-theoretic concepts [8]. However, we hold that the latter have largely failed to capture the phenomenon in a compelling and operational way (see [5]).

The behavioral literature on spatial perspective taking builds upon the distinction introduced by Flavell [51,52] between *level-1 perspective taking* (L1PT), when judging whether objects can be seen by another, which appears to operate at a rather pre-reflective level, and *level-2 perspective taking* (L2PT), which entails a more reflective and explicit simulation, deliberately attempting to imagine seeing through someone else’s eyes. In this literature, L1PT is construed as requiring one to mentally trace someone else’s line of sight, and cognitive processing times increase with the distance between the eyes/head of the agent being observed (an avatar, a doll, or a real person, depending on the study), and the target of his/her overt attention ([53]). In contrast, L2PT is construed as requiring one to reconstruct, more or less precisely, the visual appearance of the world from someone else’s perspective, and cognitive processing times increase with the angular disparity between the observer and the observed person’s line of sight ([54]). Overall, L2PT seems to be more cognitively demanding than L1PT ([26,53]). Moreover, there is evidence that L1PT can be triggered outside of cognitive control, contrary to L2PT ([55]). Further, the ability for L2PT emerges later with respect to L1PT, in terms of both human development (two-year-old children show evidence of L1PT, and L2PT seems to be fully developed later around five years of age; see [51,52,56]) and phylogeny (L2PT may be human-specific, whereas L1PT is present in the caching behavior of birds and among chimpanzees; see [57] for a review). However, there is currently a debate about whether ToM requires L2PT, when L1PT could provide a minimal ToM ([58]). This issue is certainly related to the fact that classic ToM tasks may in fact measure lower-level processes (attention orientation, face processing, etc.) that do not directly evaluate ToM ([59,60,61]) but could contribute to an implicit ToM, referred to as “submentalizing” [62], and constitute a building block of explicit ToM.

### 1.3. Projective Geometry at the Core of Consciousness

We have hypothesized that three-dimensional (3D) projective geometry is a critical ingredient that is missing in previous accounts. We hold that it is this geometry that allows us to understand and model the phenomenology and functions of consciousness and to make better sense of how they are bound together within an integrated whole structured as a subjective perspective and imbued with capacities for perspective taking. This hypothesis is at the basis of our Projective Consciousness Model (PCM) [25,63,64,65,66,67,68,69] (see also [70]).

Our primary endeavor centers on the modeling of the phenomenology and functions of conscious experience, irrespective of consciousness’s physical underpinnings (see Section 4.4 for further discussion of how the PCM may relate to NCC hypotheses). Here, our objective is not to pursue the question of how the geometry of first-person experience comes into existence. Instead, we focus on the exploration of how we can construct a quantifiable model for it, one that is amenable to empirical testing.

Projective geometry is the geometry of perspectives and points of view (see [68] for a presentation of projective geometry in a relevant context). It extends affine geometry with points at infinity, where all “parallel” lines meet, yielding a projective space. Geometrical spaces can be defined by the group that acts on them. In the case of 3D projective geometry, this group is $PGL_{3}$. Its action applies (projective) transformations that preserve the incidence structure of points, lines and planes, but not angles, and, in doing so, this action realizes changes in perspective. Group action can transform the distribution of information in space, e.g., affective and epistemic values, through group operations such as the pushforward measure (see [69] and below).

In the PCM, the action of 3D projective geometry is conceived of as an integral mechanism for systems to perform active inference [71,72,73]. Active inference is a process by which agents infer the causes of their sensations and the consequences of their actions in order to explore and exploit their environment in an optimal way. They can do so based, for instance, on the minimization of Free Energy (FE), which encodes the divergence between their expectations and their sensations or behavioral outcomes. In this context, the minimization of FE entails the maximization of affective and epistemic values and generates goal-driven and curiosity-driven actions [74]. Of note, beside FE-based approaches, active inference can be understood and modeled using other methods; all those approaches relating more generally to stochastic optimal control (see Section 2).

Through geometrically structured representations and perspective taking, the PCM operationalizes both the subjective perspective as a mechanism of information integration and the principles and functions of the global workspace from GWT (Figure 1). It does so, moreover, in a manner that captures and makes sense of core aspects of the phenomenology of consciousness. According to the model, projective geometry offers a mechanism of (conscious) access to an otherwise unconscious world model, encoding objects and their relations in a componential manner. In this context, the world model can be thought of as a homogeneous space for the 3D projective group, i.e., a space in which actions are structured by a group so that, from any point in space, all other points could potentially be reached or represented by applying the same unified principles. The hypothesis is that such a group structures the agents’ internal space of representation for active inference or more generally (stochastic) optimal control. We call the resulting space and corresponding projective transformations, as they play a core role in appraising affective and epistemic value and generating drives in relation to action planning through, for instance, the minimization of FE, the Field of Consciousness (FoC).

Interestingly, although much research still needs to be done on this topic, it also allows us to make sense of empirical distinctions such as L1PT and L2PT. Indeed, projective geometry is imbued with fundamental properties of reciprocity, related to projective duality, so that it is effectively immediate for an agent representing information within a projective space to understand relations of incidence in a pre-reflective manner [68]; this is consistent with LPT1. Likewise, the explicit action of the projective group on the space, which requires controlled projective spatial transformations, precisely corresponds to operations subsumed by the notion of LPT2. Below, our simulations of ToM build largely upon LPT2, with explicit simulations of the other’s point of view, but further work should leverage projective reciprocity and duality in a more operational manner to integrate pre-reflective mechanisms such as those described by LPT1 (see Section 4.5).

### 1.4. General Positioning

Beyond the aim of understanding, via computational modeling, how phenomenology relates to function and behavior, our approach emphasizes the role of geometrically structured representations for information integration, learning, planning, and control. It connects to geometric machine learning, topology, and data analysis, and pursues the integration of geometrical principles into active inference and reinforcement learning (RL). It shows how geometry can be leveraged in order to understand and model the dynamics of agents, building upon the duality between geometrical transformations and action. Our working hypothesis is that geometry, and more specifically 3D projective geometry, as structuring an internal subjective space via the action of a group, (1) supersedes the need for an objective representation of the environment and of the agent in its environment (e.g., exact position and metrical distances in external space) as typically required in artificial agent control and (2) plays a key role in regularizing information processing, learning, and communication, in a manner that fosters adaptability and resilience across sensorimotor contingencies for open dynamical systems and facilitates related inferences.

In what follows, we review and discuss the PCM, focusing on how projective geometry can account for the integration of perception, imagination, motivation, emotion, social cognition and action in consciousness in the context of active inference. We first present the model formally in a synthetic manner to situate its general principles. We then present and discuss key results obtained so far with the approach and introduce a new operationalization of empathy and its effect on emotion regulation and behavior based on the model. Finally, we discuss ongoing research on applications of the model in behavioral science, machine learning, and human–machine interactions, as well as perspectives for the study of the NCC.

## 2. Model

In this section, we present the model formally, from a bird’s-eye point of view, with the aim of situating our modeling framework at the highest level of generality. We have implemented such principles in specific ways in the context of specific studies, and we refer the readers to the corresponding references for details on how we did so mathematically and algorithmically [65,66,67,69]. At this point, much work remains to be performed to formulate a definitive implementation of the mathematical principles that would integrate all the components we are considering in a fully unified manner and without ad hoc solutions, which we have sometimes had to employ in order to generate simulations in specific contexts.

### 2.1. Motivation

We consider agents evolving in an environment that contains other agents. The agents plan their actions (moves) and explore their environment based on partial information obtained through observations. To do so, they model their environment through a *world model* or *state space*, and beliefs are kept about the state of the environment of the agent but also about the beliefs and action policies of other agents. Agents model the dynamics of their environment, which contains other agents, through a *generative model,* which also accounts for the consequences of their actions on the environment; it is a stochastic model of the consequences of the actions of the agents based on the current beliefs of the agents. The order of ToM of an agent quantifies how intricate the thought process of the agent is with respect to planning based on the policies of other agents, taking into account that those agents can also plan their actions based on their policies. One way to model agents with ToM is through a *interactive, partially observable Markov decision process* (I-POMDP) [75,76]. In the simplest case of one agent interacting with its environment, this reduces to a *partially observable Markov decision process* (POMDP). POMDP and active inference or the Free Energy Principle share similar generative models in cases in which agents decide on what action to do based on their current observations [73]; we discuss the similarities and dissimilarities between both approaches at the end of Section 2.3.1.

The particularity of our agents is that their world model or state space, denoted *S*, has an additional structure, that of a group that can act on the state space, denoted *G*; such a space is called a *G*-space. We will explain how a slight modification of POMDPs can account for such a structure (Section 2.4).

### 2.2. Prerequisites

Let us first recall what a group is.

**Definition 1**  (Group, §2 Chapter 1 [77])**.**
*A group is a set G with an operation: G×G→G that is associative, such that there is an element e∈G for which e.g=g for any g∈G, and any g∈G has an inverse denoted $g^{-1}$ defined as satisfying $g.g^{-1}=g^{-1}.g=e$.*

We call a group-structured (measurable) space a space provided with a (measurable) group action; we now make this statement formal.

**Definition 2** (Group-structured space, *G*-space)**.**
*S is a group-structured space for the group G when there is a map h:G×S→S denoted as h(g,s)=g.s for g∈G and s∈S, such that*
*$\left(g.g_{1}\right).s=g.\left(g_{1}.s\right)$ for all $g,g_{1}\in G$, s∈S**e.s=s, for all s∈S*
*For a given group G, this space is called a G-space.*


A homogeneous space is a *G*-space over which the group *G* acts *transitively*, i.e., from any point s∈S, any point $s^{\prime }$ can be reached via the action of an element g∈G: $g.s=s^{\prime }$.

In what follows, we assume that *S* is a topological space that is measurable (for the associated Borel σ-algebra); furthermore, we assume that *h*, the function that defines the group action on *S*, is continuous and therefore measurable.

Let us give two examples of group-structured spaces. The 3D vector space $R^{3}$ is structured by the group of invertible matrices $GL_{3}\left(R\right)$ as $GL_{3}\left(R\right)$ acts on $R^{3}$. Similarly, the projective space $P_{3}\left(R\right)$ is structured by the group of projective linear transformations $PGL(R^{3})$. Both are in fact homogeneous spaces. Let us now define the projective general linear group formally, as it is at the basis of the PCM.

**Definition 3.** 
*The 3D projective space $P_{3}\left(R\right)$ is the set of lines of $R^{4}$. Any bijective linear transformation from $R^{4}$ to $R^{4}$, i.e., any invertible 4×4 matrix denoted as M, defines a projective linear transformation in $PGL(R^{3})$.*


Homogeneous coordinates are a way to map a (dense open) subset of $P_{3}\left(R\right)$ to $R^{3}$ by remarking that when the last coordinate of $\lambda :=(\lambda_{1},\lambda_{2},\lambda_{3},\lambda_{4})$ is non-zero, then it defines the same line as $\left(\frac{\lambda_{1}}{\lambda_{4}},\frac{\lambda_{2}}{\lambda_{4}},\frac{\lambda_{3}}{\lambda_{4}},1\right)$. When expressed in homogeneous coordinates, the projective (linear) transformation can be expressed as a partial map from $R^{3}$ to $R^{3}$ defined as follows: let $\left(\lambda_{1},\lambda_{2},\lambda_{3}\right)\in R^{3}$ denote $\left(\lambda_{1},\lambda_{2},\lambda_{3},1\right)$ as $\tilde{\lambda }$; assume that the last coordinate of $M(\tilde{\lambda })$ does not vanish, i.e., $M{\left(\tilde{\lambda }\right)}_{4}\neq 0$; then, the projective transformation can be written as
(1)$$\phi \left(\lambda_{1},\lambda_{2},\lambda_{3}\right)=\left(\frac{M{\left(\tilde{\lambda }\right)}_{1}}{M{\left(\tilde{\lambda }\right)}_{4}},\frac{M{\left(\tilde{\lambda }\right)}_{2}}{M{\left(\tilde{\lambda }\right)}_{4}},\frac{M{\left(\tilde{\lambda }\right)}_{3}}{M{\left(\tilde{\lambda }\right)}_{4}}\right)$$

In the rest of the Methods section, we will present how to model agents with world models structured with a group *G*, which can be any group; *G* can be, for example, $GL_{3}\left(R\right)$, $PGL(R^{3})$ as it appears in previous work [66,67,69]—see [65] for more details on the projective general linear group—but the presented framework is not restricted to these two groups, and we are now exploring the effect of other groups on the behaviors of agents modeled with this framework.

The space of probability measures over a set *X* will be denoted as P(X). We will call a stochastic map from a space *X* to *Y*, denoted π:X→Y, a Markov kernel; more precisely, a Markov kernel π:X→Y is a (measurable) function that sends x∈X to π(.|x)∈P(Y), a probability measure on *Y*.

### 2.3. MDP, POMDP and Active Inference

**Definition 4** (Markov Decision Process: Definition 1 [78])**.**
*A Markov Decision Process is a collection 〈S,A,T,r〉 where*
*S is the set of configurations of the environment;**A is the collection of actions of the agent;**T:S×A→S is the transition probability, which captures the consequences of the action a∈A of the agent on the environment that changes from $s_{t}$ to $s_{t+1}$;**r:S×A×S→R; it is the reward function for an action a∈A and two states $\left(s,s^{\prime }\right)$ thought of as $s_{t}$ and $s_{t+1}$.*

An MDP is a model of the environment of the agent and the consequences of its actions. A policy is a prescribed way the agent acts when faced with a state *s* of its environment; it is encoded by a Markov kernel π:S→A. A policy allows us to define a probability distribution on $\prod_{t\geq 1}S\times A$ given an initial state $s_{0}$
(2)$$P_{|\pi ,s_{0}}\left(s_{k},a_{k};k\geq 1\right):=\prod_{k\geq 0}T\left(s_{k+1}|s_{k},a_{k}\right)\pi \left(a_{k}|s_{k}\right)$$

It is the distribution of planned future states and actions under the policy π. It is common to require that the agent finds a Markov kernel $\pi^{*}:S\to A$, called an optimal policy, that maximizes its utility, which is an expected sum of future rewards with horizon *t* (*t* could be *∞*):(3)$$V\left(s_{0}\right)=\max_{\pi }E_{P_{|\pi ,s_{0}}}\left[\sum_{0\leq k\leq t}\gamma^{k}r\left(s_{k+1},a_{k},s_{k}\right)\right]$$

0<γ<1 acts as a discount factor.

When the state of the environment is not known by the agent but inferred by observations, the previous formalism is changed into an extended formalism: a Partially Observable MDP.

**Definition 5** (Partially Observable Markov Decision Process [79])**.**
*A POMDP is defined as a tuple 〈S,A,T,r,O,Z〉, where 〈S,A,T,r〉 is an MDP, and*
*O is the set of possible observations.**Z is the observation kernel, Z:S×A→O, which specifies the probability of observing a particular observation given the current state and action.**r is a reward function whose domain is S×A; r:S×A→R.*

In the framework of POMDP, an agent keeps beliefs about the state space *S*, denoted as b∈P(S), that are updated through observations using Bayes’ rule. Action *a* induces a change in belief,
(4)$$T_{a}\circ b\left(s^{\prime }\right):=\sum_{s\in S}T\left(s^{\prime }|s,a\right)b\left(s\right)$$

An agent can plan the belief update induced by observation o∈O after its action *a*, defined as,
(5)$$\forall s\in S\;b_{|o,a}\left(s\right):=\frac{Z\left(o|s,a\right).T_{a}\circ b\left(s\right)}{\sum_{s\in S}Z\left(o|s,a\right).T_{a}\circ b\left(s\right)}$$

However, anticipated observations one step ahead are *theoretical* for the agent and depend on its belief about the environment; in other words, the anticipated observations are stochastic and depend on the choice of actions. A policy is a kernel π:P(S)→A that sends beliefs to actions. The distribution of anticipated observations is then given by
(6)$$P_{O_{1}}^{b}\left(o\right):=\sum_{a\in A}\left(\sum_{s^{\prime }\in S}Z\left(o|s^{\prime },a\right)\sum_{s\in S}T\left(s^{\prime }|s,a\right)b\left(s\right)\right)\pi \left(a|b\right)$$

In order to introduce the usual utility function that an agent with partial observations wants to maximize, let us now show that POMDPs are particular MDPs. A POMDP can be reformulated as an MDP where the state of the environment is replaced by the space of possible beliefs about the environment; such a process is called a *belief* MDP. The following is a dictionary:$\tilde{S}:=P\left(S\right)$$\tilde{A}:=A$$\tilde{T}:\tilde{S}\times \tilde{A}\to \tilde{S}$ is defined as
(7)$$\tilde{T}\left(b^{\prime }|b,a\right)=\sum_{o\in O}P_{O_{1}}^{b}\left(o\right)1\left[b^{\prime }=b_{|o,a}\right]$$$\tilde{r}:\tilde{S}\times A\to R$ is defined as
$$\tilde{r}\left(b\right)=\sum_{s\in S}b\left(s\right)r\left(s,a\right)$$

The utility of the POMDP is the utility of the belief MDP defined by the dictionary.

#### 2.3.1. Relation between POMDP and the Free Energy Principle

POMDPs, active inference, and the Free Energy Principle share similar generative models; in particular, in our previous work [66,67,69], we considered such models when agents decide on what action to take based on their current observations. One (minor) difference between both frameworks is the objective function of the agent. For POMDP, in the context of stochastic optimal control theory, it is encoded by a sum of expected rewards (value function), and for the free energy principle, it is a probabilistic version of this function that is considered (duality between Bayesian estimation and stochastic optimal control [80,81]); however, both formalisms share many similarities [73]. In the rest of the article, we refer to how the value function relates to the states of agents as the “affective value” of the actual or anticipated state. An important difference between the two frameworks is that in POMDPs, belief updates are performed through the exact application of Bayes’ rule, while in active inference it is through an approximation of such a rule (approximate variational inference). We chose to present our framework as a specialization of POMDPs as POMDPs constitute a standard way to model agents interacting with their environments. However, as our modification only concerns the space on which the beliefs of the agent are kept, our approach can be transcribed in terms of the Free Energy Principle.

### 2.4. POMDP with Group-Structured State Space

We will call an MDP with group-structured state space an MDP where the state space *S* is a *G*-space, for some group *G*, and a subset of the set of actions is the group *G*.

**Definition 6** (MDP and POMDP with group-structured state space)**.**
*An MDP with a group-structured state space is a tuple 〈S,A,T,r,G〉 where G is a group and 〈S,A,T,r〉 is an MDP that satisfies the following properties:*
*S is a G-space**G is a subset of the set of actions A,**For all g∈G, $T\left(s^{\prime }|s,g\right)=1\left[s^{\prime }=g.s\right]$**A POMDP with a group-structured state space is a tuple 〈S,A,T,r,O,Z,G〉 where 〈S,A,T,r,G〉 is a group-structured MDP (structured by G) and 〈S,A,T,r,O,Z〉 is a POMDP.*

**Remark 1.** 
*One should note that the action of an element of the group g∈G on g:S→S induces an action on the beliefs over S, defined as, for any b∈P(S), A is a measurable subset of S,*

(8)
$$\forall A\subseteq S,g_{*}b\left(A\right):=b\left(g^{-1}\left(A\right)\right)$$


*However, following the same procedure as the one that transforms a POMDP into a (belief) MDP does not make a POMDP with a group-structured state space into a (belief) MDP with a group-structured state space: $\tilde{T}$ is a stochastic map and not a deterministic map; therefore, it cannot come from the action of a group on the space of beliefs.*


We think of *G* as all the actions the agent can perform that do not change its environment but change the way it perceives its environment. For example, in our work up to now, *G* contains the movements that the agent can make in its environment; these movements change the representation the agent has of its environment through a change in the egocentric chart. Our formulation is to be opposed to a state space equipped with a reference frame global to all entities in the environment, the agent included; in this latter formulation, moves of the agent only change its position in the environment, and to be accounted for, it requires the agent to model its own configuration in the environment. Of course, higher-level cognition involves such a configuration, which is an extension of what we describe in this section, but here, we wish to focus on the most basic mechanisms of interest. Our proposition is to encode the actions of the agent that only change its perception of its environment as a transformation of the state space and disregard the configuration of the state space for entities that are not the agent. In particular, we believe that such an approach could adequately accommodate evolving the perceptual skills of the agent, for instance, through the addition of new sensors into the same theoretical framework.

We propose that imbuing world models with a “geometric” structure, given by a group, is one way to capture different perception schemes of agents. In particular, in [69], we explore how changing the geometric structure of a state space, namely the group *G* acting on *S*, impacts the behavior of an agent; we consider a reward based on the relative entropy
(9)$$\mathrm{DKL}\left(b^{\prime }∥b\right)=\sum_{s\in S}b^{\prime }\left(s\right)\ln\frac{b^{\prime }\left(s\right)}{b(s)}$$
and horizon T=1; the associated objective function corresponds to the “epistemic value” [74].

When agents must model other agents and their beliefs, one can specify further POMDPs into interactive I-POMDPs [75]. We attempted to mimic such a formalism in an approximate formulation proposed in [66,67]. Modeling interacting agents with group-structured state space can be performed with I-POMDPs by requiring that every agent simulates other agents that have group-structured state spaces. It is one way to accommodate both I-POMDPs and group-structured state spaces.

To conclude, according to the formulation of the PCM we introduced above, looking for a working definition of consciousness to better situate our approach on the map of possible classes of models at a very high level of generality, we can say that conscious agents can be modeled using an Interactive Partially Observable Markov Decision Process whose state space is structured by the action of the Projective Group.

## 3. Results

In this section, we review published results obtained based on the principles of the PCM. We also show novel preliminary results derived from applying the model to simulating processes related to empathy, emotion regulation, and its role in the control of approach–avoidance behaviors.

### 3.1. Perceptual Illusions

In [65], we proposed an initial version of the model that focused on visual perception, aiming to account for perceptual illusions, in particular the Moon Illusion, whereby when the Moon is low on the horizon, its size appears bigger than when it is high in the sky (Figure 2). The model’s predictions were validated in a virtual reality experiment by comparing simulations of the Moon’s apparent size as a function of its elevation, with psychophysical estimations of relative apparent size by human participants. In the same contribution, we also accounted for the Ames Room Illusion, which uses forced perspective on a geometrically deformed room to induce the illusion that two persons standing against a back wall appear to have radically different sizes. See also [25] for initial accounts of other illusions such as the Necker Cube, Heautoscopy (the experience of the perceptual reduplication of one’s own body accompanied by an ambiguous sense of self-location), and Out-of-Body Experiences. In all these cases, the illusions were conceived of as arising from the calibration of a 3D projective frame under the minimization of free energy (FE). They resulted from the way information is conditioned by priors and accessed in consciousness according to the model, generating a posterior representation structured by 3D projective geometry, corresponding to a perceptual content within the subjective perspective.

### 3.2. Imagination, Emotion, Drives, Social Cognition, and Adaptive Behaviors

In [66,67], we introduced a more encompassing model and software implementation to study how the model could account for adaptive and maladaptive social behaviors through mechanisms of perspective-taking in robots and virtual human agents (Figure 3).

A pivotal operation was the quantification of affective and epistemic values as a function of projective transformations associated with possible actions, e.g., moving in a certain direction. Projective transformations induce a magnification (or shrinking) of information in the space equivalent to the effect that approach (or avoidance) behaviors have on perception, according to the model. The transformed information was integrated spatially to compute affective and epistemic values of actions; these related to how much of the FoC that information would occupy as a function of action (see [66,67] for technical details). The rationale for the relationship between projective transformations, as magnifying or shrinking the apparent size of information in space, and affective or epistemic value was that agents that want or are curious about something should approach that thing, effectively making it bigger in their FoC, while agents that do not want or are uninterested in something should avoid that thing, effectively making it smaller in their FoC. The resulting relationships between affective value and the distance *z* of an object or another agent from the agent of interest (about which the latter agent had prior positive or negative preferences *p*) yielded a law approximating 1/z. This was consistent with psychophysical empirical findings about the intensity of felt negative emotions as a function of the distance of threatening stimuli [82]. Affective and epistemic values of anticipated actions were then used to control parametric probability distributions and compute the Kullback–Leibler divergence (DKL) between those distributions and ideal distributions representing goals as an approximation of FE. The minimization of FE induced corresponding affective and epistemic drives to approach or to avoid objects and other agents. Using the same basic operations, agents could simulate each other’s FoC, making inferences about the preferences and uncertainties of others based on emotional expressions and spatial configurations and predicting each other’s behaviors accordingly. This enabled agents to further minimize their expected FE by taking into account others’ expected behaviors. Mechanisms of social influence, related to normative (conformism) or informational (acquisition of new interests) influences [83], were also implemented by manipulating the weight of the expected FE attributed to another agent in the computation of an agent’s own FE, or the update of the agent’s own prior preferences as a function of preferences attributed to the other.

The action of the projective group on affective and epistemic values directly contributed to maximizing expectation satisfaction and information gain in the agents, resulting in different strategies of action combining exploration and exploitation. Through emotional expression and projective geometry, social agents could communicate and understand each other, enabling them to infer key information about their environments. They could simulate each others’ minds recursively, relying on their spatial and affective behaviors, in a manner that fostered the transfer of information localization, i.e., the operation, which relates to attention, of restricting relevant information to certain regions of space in the agents’ internal representation.

On this basis, in [66], we could generate adaptive and maladaptive behaviors in robots. This work is relevant to developmental and clinical psychology, specifically in relation to the ability to be resilient through imaginary projections when confronted with obstacles; social-approach and joint-attention behaviors; the ability to take advantage of false beliefs attributed to others; avoidance behaviors typical of social anxiety disorders; and restricted interests, as observed in autism spectrum disorders.

In Figure 3a, we show simulations performed for a non-verbal version of the classic Sally and Anne Test [84,85], operationalizing an objectivization of the ability to take advantage of false beliefs attributed to others. Two robots (Anki Cozmos) were used, with robot S, the robot of interest (the subject), and robot O, the other, along with two objects, cubes c1 and c2. This simulation aimed at assessing the ability of S to take advantage of another agent O’s false beliefs. We operationalized Sally and Anne Test for our non-verbal context, using a competitive situation between agents generating a conflict between approach and avoidance for S. Cube c2 was associated with positive prior preferences for both agents. Cube c1 was neutral. Both agents believed that the other had positive preferences for c2. S disliked O and believed O disliked S, but O was neutral about S. Even though approaching c2 would minimize FE for S in isolation, the prediction by S that O would approach c2 made S tend to avoid c2 in order to avoid O. The simulation was divided into two phases: at iteration 30, S and O were re-positioned at their initial location, and the locations of c1 and c2 were switched. Two conditions were contrasted. *Condition tb*: O had true belief about the location of c2 at all times. Both S and O could witness the switching of the cubes and maintain true beliefs about their location, and they could understand that the other agent had true beliefs about it. *Condition fb*: in phase 2, O had false belief about the location of c2, as, before switching between cubes c1 and c2, O was rotated so it could not witness the switching. S, being a witness of that contingency, inferred that O would hold false beliefs about the location of c2. In *Condition tb*, robot S could not approach its preferred cube c2 as it expected robot O to approach it and had to move away from the scene to minimize its FE (see (T2) in Figure 3a). In *Condition fb* (phase 2), after the locations of c1 and c2 were switched, robot S could approach its preferred cube c2, as it rightfully expected robot O to approach the wrong cube, thus further minimizing its FE.

In [67] (see Figure 3b), we used two virtual humans, a subject S and another agent O, starting on opposite sides of a small building (middle rectangle in the figure top-tier), competing for access to vending machines VM1 (right) and VM2 (left) on opposite ends of a gas station, with different intrinsic values (one being better than the other). Both O and S liked (positive signs) VM1 and VM2 but preferred VM1 (longer arrow). S disliked O and believed that O disliked S (negative signs) and thus would try to avoid S to minimize its FE. In fact, O liked S. The aim of this simulation was to assess the ability of S to infer the order of O’s Theory of Mind (ToM) and its preference toward S in order to optimize the outcome, i.e., to have minimal FE at the end. Agents could not see each other except when arriving together near a vending machine. ToM of order 0 (ToM-0) corresponded to no ToM, ToM of order 1 (ToM-1) to the simulation of the other as performing ToM-0, ToM of order 2 (ToM-2) to the simulation of the other as performing ToM-1, and so on, up to order 3 in this publication. When agents would run into each other at a vending machine, they could use their observations of approach–avoidance behaviors and emotional expressions (negative versus positive reactions to running into another) as evidence to update the preferences they attributed to the other. Agents demonstrated a variety of behaviors as a function of initial conditions that were consistent with behaviors we would expect in humans performing a similar task. The simulation was divided into two trials. In the example shown in Figure 3b, in trial 1, S initially assumed wrongly that O was performing ToM of order 0 (ToM-0), i.e., no ToM, whereas O was actually performing ToM of order 2 (ToM-2). O correctly predicted that S would go to VM2 in order to avoid O. Since O liked S, O went to VM2. Both S and O ended up finding themselves at VM2. S could then use sensory evidence to revise its priors. In Trial 2, S selected ToM of order 3 (ToM-3), correctly attributing ToM-2 to O and positive preferences (*p* = 0.8) of O toward S. S then chose to go to VM1, both maximizing reward in terms of VM and avoiding O, which resulted in minimal FE.

In [69], we further proved theoretical results demonstrating that changing the group that structures the internal world model of the agents influences their curiosity-driven exploratory behavior. We compared the action of the Euclidean Group to that of the Projective Group on the computation and maximization of epistemic value and on the ensuing behaviors of exploration in a simple search task. Only the Projective Group induced behaviors of approach toward the uncertain location of an object of interest due to its effect of magnification on information and how such an effect influenced epistemic value and induced a drive under FE minimization. This result further suggests that projective geometry has unique properties for supporting information integration, valuation, and action planning in adaptive systems.

### 3.3. Application to Empathy, the Regulation of Emotion Expression, and the Control of Approach–Avoidance Behaviors

Here, we aim to show, in a preliminary manner, how the PCM can be further employed to operationalize mechanisms of empathy and affective processing to control behaviors, using new simulations and building upon the previous work and software presented above. These are preliminary results and are to be taken as an indication that, overall, the model behaves as expected; but many details still need to be worked out before the approach can be used in experimental settings. We were interested in the relationships between empathetic processes, the regulation of emotion expression under active inference, and the control of approach–avoidance behaviors. The goal here was only to assess whether the model could simulate these types of mechanisms and phenomena as a proof of concept. However, we believe it is not trivial that the algorithm can produce such effects just by adding the expression of emotion in the repertory of actions subject to active inference, e.g., assessing the impact on FE of choosing to smile or frown. The modeling approach is consistent with simulation theories of empathy, social perspective taking, affective learning, and emotion regulation [40,41]. Under these principles, humans use their own cognitive and affective apparatus to imagine themselves in the position of others in order to simulate their subjective experience and infer their likely behaviors. This process in turn can be used to control one’s own behavior. Thus, we are leveraging LPT2 mechanisms of ToM (see Section 1.3). In particular, we wanted to incorporate emotional expression under voluntary control into the repertory of behaviors used by agents for FE minimization as part of active inference. We used previously published algorithms and corresponding software from [66,67] and added this feature. We considered how simulating the FoC of another agent performing ToM could lead an agent to express emotions that are opposite to those it actually undergoes in relation to the other agent because of social influence of, e.g., conformism (see [66] for technical details about the operationalization of this concept). We simulated a simple dyadic situation in which two agents, A0 and A1, faced each other, across several combinations of parameters (Figure 4). For instance, in one combination of parameters, A0 did not like A1. In all conditions, A1 was prosocial (it liked to meet new agents), so it had a positive prior about A0. A0 was aware of that prior. Then, in the example considered, A0 had two choices, expressing the negative emotion it was experiencing or a positive emotion against its own preferences. Because of normative social influences, A0 took into account, in the computation of its own FE, the expected FE it attributed to A1 through ToM (related to the inferred affective state of the other agent) as a function of its choice of emotion expression. Expressing a negative emotion would increase FE in A1, while expressing a positive emotion would decrease it. As a result, A0 chose to express a positive emotion to minimize its own FE.

Furthermore, we considered both voluntary and involuntary aspects of emotional expression using the principles and virtual humans implemented in [86]. Facial expressions of virtual humans had two main components: a musculoskeletal component, which was operationalized using action units (AUs) ([87]), controlling features such as smiling or frowning, and a physiological component, related to the Autonomic Nervous System (ANS), with two subprocesses related to the tone of the sympathetic versus parasympathetic branches of the ANS. The ANS component controlled features such as pupil diameter, skin tone (related to blood surface capillaries’ perfusion), and sweating. High parasympathetic tone entailed pupil contraction, reddish skin tone, and no sweating, whereas high sympathetic tone entailed pupil dilatation, pale skin tone and sweating. The ANS component was assumed to be involuntary and hard to control. We simulated agent A1 in two conditions (Figure 5): (1) with minimal sensitivity to the ANS features expressed by A0 versus (2) with maximal sensitivity to those features (we used a simple weighted average as a first implementation of the sensitivity function). A0 expressed voluntary positive emotions through the AU component to minimize its own FE, even though it disliked A1. Thus, it also involuntarily expressed its negative felt emotion through increased pupil diameter, paleness and sweat. A0 was assumed to be at a fixed position so that it would not move away when seeing A1. In condition (1), A1 was only sensitive to the AU component and thus wrongly inferred that A0 was happy to be approached by A1. As a result, A1 approached A0, making A0 very uncomfortable. In condition (2), A1 was sensitive to the ANS component and thus correctly inferred that A0 would not be happy to be approached by A1. As a result, A1 moved away from A0.

## 4. Discussion

Although much theoretical and experimental work remains to be done, the PCM offers a powerful account of the integrative and functional role often ascribed to consciousness that is consistent with core aspects of its phenomenology. It accomplishes this by bringing forth the fundamental structuring role of 3D projective geometry in information processing and optimal planning in the context of active inference or more generally optimal stochastic planning. Projective geometry appears to be able to operate as an internal subjective perspective that acts on a variety of types of information to relate multiple cognitive functions and processes into a global workspace. In this last section, we wish to briefly discuss related perspectives and ongoing axes of research that we intend to pursue based on the PCM.

### 4.1. Behavioral Science

One of the motivations of our approach is to study how consciousness influences behaviors and how behaviors influence consciousness. The PCM has the advantage of offering an operational framework that can be implemented for empirical research based on states and observable behaviors that can be quantified in humans. This can be performed independently from any hypothesis about the NCC. Mathematical models can be implemented computationally in a precise manner. Hypotheses can be formally expressed and tested by comparing simulations and human behaviors. In particular, we are interested in combining the model with virtual reality as a space of interaction and observation. We started carrying this out in our work on the Moon Illusion [65] and developed a simulation framework in our work on ToM in virtual humans and its current extension for studying the relationships between empathy, emotional regulation, and behaviors of approach and avoidance, which provides a groundwork for future research on social cognition and more complex social behaviors. The overarching goal is to design tools for standardized, model-based psychometric assessments of social cognition in virtual reality, leveraging the PCM in order to enable inference-driven interactions between artificial agents and real humans, in different conditions and across different populations, including clinical populations.

### 4.2. Machine Learning

Another axis of research we are actively pursuing is to investigate the potential advantages of using geometrically structured representations over more classical machine learning (ML) approaches, such as state reinforcement learning (SRL) [88] (see [69] for the background and rationale). We are now working on deriving theorems for and implementations of what we call Perspective Neural Networks (PNNs), in which geometrical frames are associated with internal layers in order to regularize network inferences. We are currently considering two main directions of research in this context. The first one pertains to addressing how geometry could play a role in attentional mechanisms for optimizing learning and inference. The effects of the relative magnification and shrinking of information due to the action of the Projective Group are notably of interest for spatial attention. The second one concerns domain adaptation and learning transfer across input modalities, such as transfer from visual information to haptic information for inference and object recognition. The representation learning of the action of geometrical frames within an internal global workspace could mediate the control of inferences across modalities and facilitate learning transfer [89].

### 4.3. Human–Machine Interfaces and Interactions

A third axis of research is to further develop our models and implementations, including through the integration of ML mechanisms such as PNN, in order to design robotic and virtual agents that will more naturally interact with humans and that will be more explainable (pre hocand post hoc) by humans than systems based purely on models such as deep learning and deep RL. Geometrically structured world models lend themselves well to intuitive, shared representations between different agents. The tools we have developed in our line of research could be employed to design artificial agents in a way that makes their internal representations intrinsically explainable, which is an important goal of XAI (explainable AI). By accessing the geometric representation of the agent, a human subject could highlight key features the agent should focus on in order to improve learning, for example. We expect such methods to be highly useful, including when interacting with agents that have to accomplish tasks that are highly unnatural for humans (e.g., too small a scale or too large a scale), so that common geometric representations would build a bridge between humans and machines, enabling a common lexicon, so to speak. Furthermore, we expect that interacting with agents following the PCM principles will render those interactions more human-like and natural to users, as could be assessed through user experience experiments. Likewise, the PCM includes explicit parameters and states that have a direct psychological interpretation and can thus be used to explain (and control) the behaviors of agents.

### 4.4. The Neural Correlates of Consciousness

Another axis of research, which is at this point more remote than the previous ones in our agenda, is to test hypotheses about the NCC based on the PCM. For instance, one of the predictions of the PCM is that consciousness accesses and processes information by bringing it into a projective space and transforming it through the action of the projective group [65]. We have proposed some preliminary hypotheses about the anatomo-functional underpinning of the process (see Section 4.1 in [25,65]).

In [25], we predicted that the brain embeds two main engines that are coupled: (1) a higher-level inference engine integrating systems concerned with homeostasis, emotion, memory, language and executive functions, or, more generally, personal relevance for agents; and (2) a lower-level (sensorimotor) projective geometry engine, concerned with multisensory integration and motor programming and representing the world and the body in space. We hypothesized that the inference engine involves anterior cortical and subcortical systems, including limbic and non-limbic frontal and temporal association cortices, the amygdala and the hippocampus, and that the projective geometry engine involves posterior temporal–parietal–occipital, modal and multimodal sensory systems, in particular parietal systems, integrating exteroceptive, proprioceptive and interoceptive processing, but also frontal premotor regions.

Spatial memory and affective or personal relevance processing [90] are tightly related in the brain, e.g., through the interactions between the hippocampus [91] and amygdala [90], and more generally, through interactions between regions of the so-called Default Mode Network (DMN), including medial temporal systems [92]. On the other hand, occipital, posterior temporal and parietal regions are strongly related to spatial transformations and processing [93,94,95,96,97]).

When retrieving autobiographical memory, one can adopt an internal perspective, that is, a first-person perspective (1PP), or an external/observer third-person perspective on oneself (3PP). Interestingly, the adopted perspective (whether internal or external) during memory retrieval depends on the nature of the emotion associated with the event [98]. Emotional events are more likely to be remembered through an internal 1PP than through an external 3PP. Reciprocally, the viewpoint used during autobiographical memory retrieval can influence how we perceive the emotional intensity of memories so that memories associated with internal perspectives are more emotionally intense than memories associated with external perspectives [99] (see also [100]). Among the brain regions supporting changes in emotion due to shifting perspective during autobiographical memory retrieval, there are the amygdala and the precuneus [101]. The amygdala supports the emotional experience associated with the retrieval of personal memories [102]. The precuneus is an associative region within the human posteromedial cortex [103] involved in visuo-spatial perspective taking [104] and supposedly a core system for the sense of self [27,105] (see also [106]).

The still ongoing debate in cognitive neuroscience regarding whether conscious access and experience require frontal–parietal interactions or solely activity in posterior cortices [2,13] might be partially driven by a focus on different aspects of consciousness relating to the division into two main engines in our hypothesis.

However, beyond such general anatomic–functional hypotheses, further developments are required to precisely formulate quantitative hypotheses that could be operationalized in order to test them using electrophysiological and neuroimaging methods with a high degree of sensibility and specificity. Generally speaking, we could design neuroimaging experiments to probe the NCC based on parametric manipulations aimed at isolating neural systems and interactions consistent with 3D projective geometrical operations mediating the minimization of FE.

Likewise, recent work [107] suggests that the geometry of the anatomical organization of the brain may constrain the propagation and interaction of its electrical fields. However, given our current knowledge, the geometry and functional processes underlying the PCM cannot be directly related to such principles in any rigorous or meaningful way, that is, using mathematics, yet. This applies to other proposals that might turn out to be relevant, e.g., the Temporo-spatial Theory of Consciousness (TTC) [108], the hypothesis that scale-free activity in the brain may underpin the subjective point of view [109], or the anatomo-functional hypotheses derived from studies of recovery from general anesthesia [110].

Such current limitations are true, more generally, about any neural model at this point. Our view (at least that of D. Rudrauf) is that the process we are considering involves a form of “virtualization” that we expect to be quite indirectly related to the anatomical and functional processes we can currently observe and model in the brain [6].

### 4.5. Pre-Reflective Self-Consciousness

Although reflective deliberation is a central aspect of conscious processing for adaptation to the world (e.g., [68,111]), one outstanding issue concerning theories of consciousness is to account for pre-reflective self-consciousness (PRSC), i.e., the property of consciousness to be pre-reflectively conscious of itself. This is both a highly debated and often unsatisfactorily posed topic in the field. We have speculated that there might be a deep relationship between the perspectival character of consciousness as governed by 3D projective geometry and PRSC [68]. Our hypothesis is that the fundamental and unique type of duality that is at the core of projective geometry might account for the pervasive yet elusive experience of feeling aimed at (or looked at) when we aim at (or look at) something (I look into the sky and sometimes it feels in a way like the sky is looking back at me). Such reciprocity between the observed and the observer arises naturally from duality in projective geometry. The question of how such properties and phenomena relate to cognition and behavior remains to be addressed. Generally speaking, one might hypothesize that it entails a basic form of built-in intersubjectivity and makes us always already prepared to take our experience as if it were viewed by somebody or something else. Such a feature might be expected to facilitate ToM and related behaviors for instance. However, the mathematical operationalization and algorithmic integration of such a mechanism into our model is not a trivial issue; it should be the topic of future work.

## Figures and Tables

**Figure 1 brainsci-13-01435-f001:**
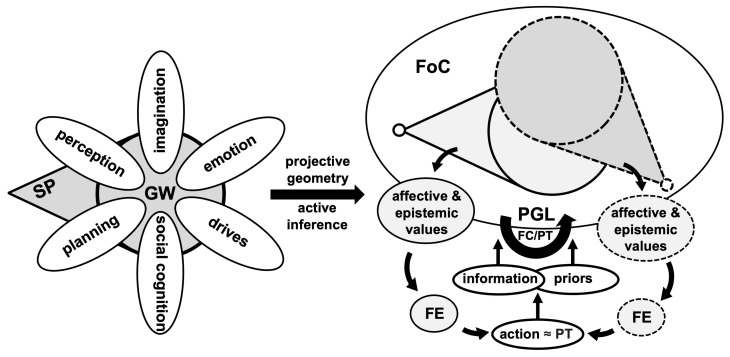
**Modeling approach: from metaphors to computation.** (**Left Tier**) Two principles to be combined: A Global Workspace (GW), integrating and processing multiple sources and types of information and priors, and a Subjective Perspective (SP). (**Right Tier**) Field of Consciousness (FoC), projective geometry and active inference, as a GW through a SP. The FoC is structured by a 3D projective space, undergoing transformations through the action of the projective group (PGL) for perspective taking (PT). Each possible perspective is associated with affective and epistemic values depending on the distribution of information in the space, with the values themselves yielding a value of FE. The projective transformation associated with the lowest expected FE is selected, providing the agent with a model for its actions (moving so as to adopt the perspective minimizing the FE). The approach is based on the duality between PT and actual or imagined actions in ambient space. At the lowest level of processing, the FoC is calibrated (FC) to select the specific projective framing of information in the projective space (which modulates the precise representation and perception of information in space). This process underlies conscious access to information and is the basis for multiple perceptual illusions.

**Figure 2 brainsci-13-01435-f002:**
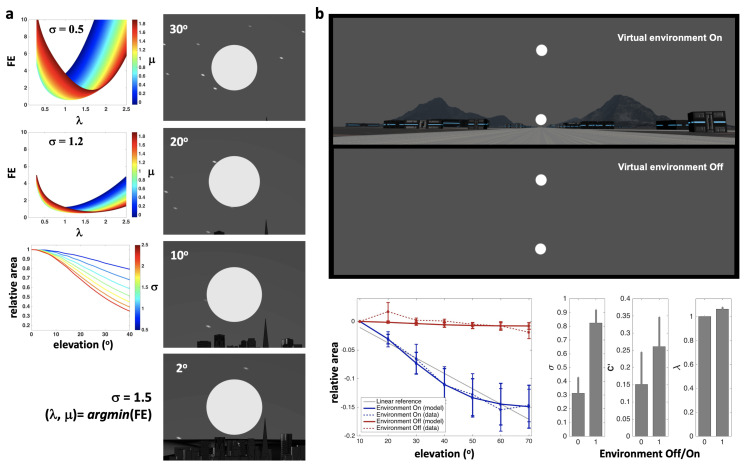
**Perceptual illusions: the Moon Illusion.** (**a**) Simulations. *Left-Tier*: Charts of relations between parameters in the model. *Top and Middle*: FE as a function of projective parameters λ, μ, and σ. The FE function is strictly convex, guaranteeing a unique solution. *Bottom*: Relative area of the perceived Moon as a function of elevation (in degrees) and σ, showing a range of possible magnifications. *Right Tier*: Rendering of a world model (including the Moon at projective infinity) in a projective 3-space as a function of elevation (in degrees), whose calibration resulted from minimizing FE. (**b**) Validation in virtual reality. *Top Tier*: Virtual Reality (VR) scenes for two conditions, environment On versus Off, displaying a reference moon (near the horizon) and a target moon at a given elevation. The task for the participants was to adjust the perceived size of the reference moon to make it match that of the target moons at various elevations. *Bottom Tier*: Result charts. (Error bars are standard errors). *Left*: Between-participant average perceived relative area as a function of elevation and condition. With the environment cues On (blue) versus Off (red), on average, the empirically perceived areas (dashed curves) decreased versus did not decrease with elevation, demonstrating an effective Moon Illusion in VR. On average, the PCM predicted the observed nonlinear curves (continuous lines) with a better predictive power than a linear model (grey line). *Right*: Average (and variability of) PCM parameters, σ, $C^{\prime }$, and λ estimated from empirical data, as a function of the presence (1) or absence (0) of environmental information. The estimated projective parameters, which control the calibration of participants’ FoC according to the PCM, could offer model-based psychometric metrics, representing features of the detailed projective structure of the participants’ individual consciousness. See [65].

**Figure 3 brainsci-13-01435-f003:**
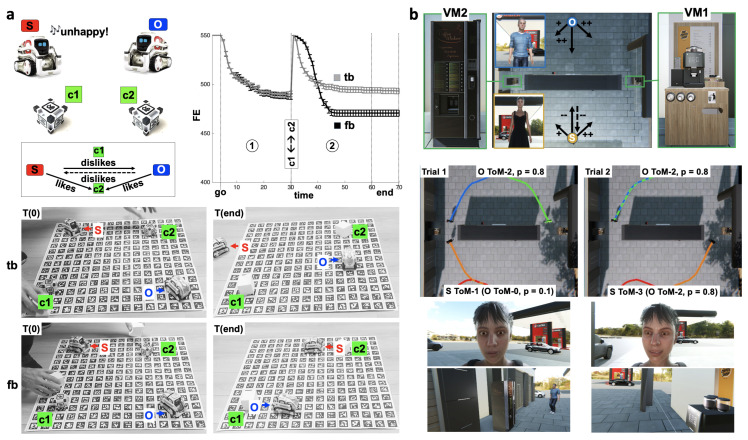
**Simulations of social–affective agents.** (**a**) Robotic context. *Top Tier*. *Left*: Setup. Two robots (Anki Cozmos) and two objects (cubes). Robot S is the subject, i.e., the robot of interest. In the small bottom chart, arrows indicate whether an agent likes or dislikes a cube or an agent. The dashed arrow from O to S indicates beliefs held by S about O. The absence of an arrow implies neutral preference. *Right*: Chart of the FE of S as a function of time (iterations), for the two conditions tb (true beliefs) and fb (false beliefs). Average FE across trials (error bars: standard errors). *Bottom Tier*. Illustration of the situation with actual robots. Snapshots are shown for two time points: T(0) and T(end). T(0) corresponds to the beginning of phase 2. *Upper row*: Condition tb phase 2. *Lower row*: Condition fb phase 2. (**b**) Virtual humans. *Top Tier*: Setup. Arrows from circles marking the initial position of S and O indicate fixed initial prior preferences towards entities. *Middle Tier*: Results. Views from above of virtual environment for Trial 1 (left) and Trial 2 (right). Trajectories: orange traces are S, blue traces are O, green traces are predictions about O according to S, and red traces are predictions about S according to O. *Bottom Tier*: *Top*, face close-up of S as a female virtual human; *Bottom*, first-person perspective of S on O (male virtual human).

**Figure 4 brainsci-13-01435-f004:**
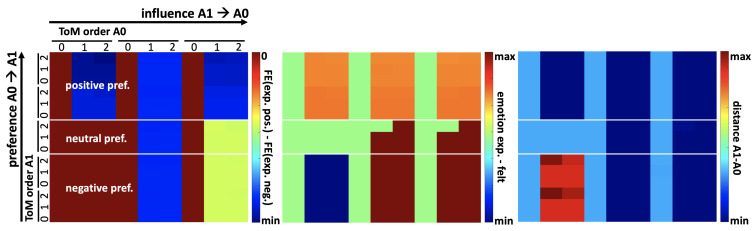
**Overall relationships between FE, emotion expression and approach–avoidance behaviors as a function of parameters.** Simulations of dyadic interactions between A0 and A1. Several combinations of parameters were varied: the preference of A0 toward A1 (negative, neutral or positive); the level of social influence A1 had on A0 (from none to high; the influence of A0 on A1 was assumed to be high); and the order of ToM used by A1 and A0 to perform active inference (from ToM-0 to ToM-2). *Left chart*: The contribution to FE of expressing a positive emotion minus the contribution to FE of expressing a negative emotion (see color bar). When expressing a positive emotion is advantageous compared to expressing a negative emotion, that dependent variable entails negative values. For higher levels of social influence of A1 on A0, and for negative or neutral preferences of A0 toward A1, as well as at least a ToM order of 1 used by A0, expressing a positive emotion yielded a lower amount of FE in A0, as expected. For positive preferences of A0 toward A1, even in the absence of a direct social influence, expressing a positive emotion was still advantageous as A0 anticipated that it would drive A1 to approach A0, which would minimize A0’s FE in this condition. *Middle chart*: Emotion expressed (from negative to positive) minus emotion felt (from negative to positive). In the absence of a social influence of A1 on A0, and for negative preference of A0 toward A1, as well as at least a ToM order of 1 used by A0, the emotion felt by A0 was more negative than its expressed emotion, but both were negative. For positive preferences, the emotion expressed by A0 was more positive than the positive emotion felt by A0. For higher levels of social influence of A1 on A0, the emotion expressed by A0 was more positive than the emotion felt by A0 (in particular for negative preferences of A0 toward A1). This effect might indicate a need to find better solutions for the normalization of parameters in the implementation (more generally, the generative model of emotion expression needs serious developments beyond the simplistic solutions we used for practical reasons). *Right chart*: Distance between A1 and A0. In the absence of the social influence of A1 on A0, and for negative preference of A0 toward A1, as well as at least a ToM order of 1 used by A0, A1 moved away, as A0 expressed a negative emotion. Otherwise, in general, A1 tended to approach A0, thus reducing the distance.

**Figure 5 brainsci-13-01435-f005:**
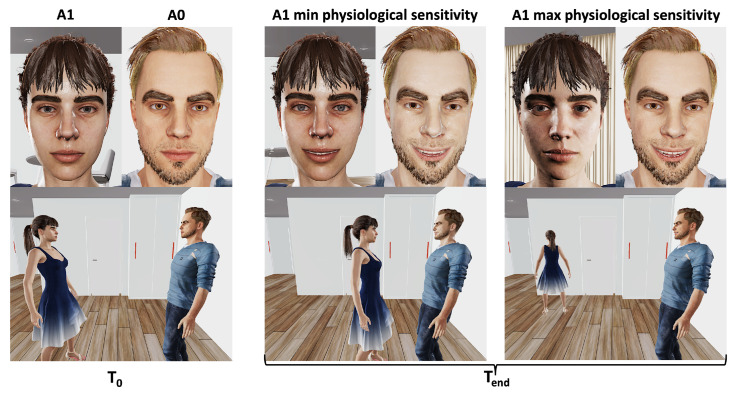
**Impact of sensitivity to involuntary emotion expression on the agents’ dynamics.** Perception of voluntary versus involuntary aspects of emotional expression and approach–avoidance behaviors. $T_{0}$ corresponds to the initial setup in both condition (1), in which A1 was only sensitive to the AU component, and condition (2), in which A1 was sensitive to the ANS component. $T_{end}$ corresponded to the end state of the simulations for both conditions. See text.

## Data Availability

Not applicable.

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
