# Peer review of "The Projective Consciousness Model: Projective Geometry at the Core of Consciousness and the Integration of Perception, Imagination, Motivation, Emotion, Social Cognition and Action"

_brainsci, 2023, doi:10.3390/brainsci13101435_

Round 1

Reviewer 1 Report

I agree with the observations in the paper “A wealth of proposals of varying degrees of precision and heuristic value have flourished over the years. Yet there remains no consensus about which contender might be most promising.” I also agree that computational modeling can better be used to understand how consciousness integrates “perception, imagination, emotion and action programming for adaptive decision making.” The author’s effort to discuss how PCM enables us to model and simulate group-structured drives, in the context of social cognition and to understand mechanisms underpinning empathy, emotion expression and regulation, and approach-avoidance behaviors. The authors have extensive references and the discussion is very informative in seeing how models and simulations can help in understanding diverse aspects of a very complex subject.

The authors also discuss current models IIT and GWT and their relevance to understanding consciousness.

While I recommend the publication as is, I would like to point out a few observations about the theories of consciousness in light of the general theory of information (GTI) articulated by the Late Prof. Mark Burgin, the works of Stanislas Dehaene, and Antonio Damasio.

According to Stanislas Dehaene [1], “conscious perception transforms incoming information into an internal code that allows it to be processed in unique ways.” It fulfills an operational role. “Consciousness implies a natural division of labor. In the basement, an army of unconscious workers does the exhausting work, sifting through piles of data. Meanwhile, at the top, a select board of executives, examining only a brief of the situation, slowly makes conscious decisions.” He goes on to say “In fact, consciousness supports a number of specific operations that cannot unfold unconsciously. Subliminal information is evanescent, but conscious information is stable—we can hang our hat on to it as long as we wish. Consciousness also compresses the incoming information, reducing an immense stream of sense data to a small set of carefully selected bite-size symbols. The sampled information then can be routed to another processing stage, allowing us to perform carefully controlled chains of operations, much like a serial computer. This broadcasting function of consciousness is essential. In humans, it is greatly enhanced by language, which lets us distribute our conscious thoughts across the social network.”

According to Antonio Damasio [2], “Conscious deliberation, under the guidance of a robust self, built on an organized autobiography and a defined identity, is a major consequence of consciousness, precisely the kind of achievement that gives the lie to the notion that consciousness is a useless epiphenomenon, a decoration without which brains would run the life-management business just as effectively and without the hassle. We cannot run our kind of life, in the physical and social environments that have become the human habitat, without reflective, conscious deliberation. But it is also the case that the products of conscious deliberation are significantly limited by a large array of nonconscious biases, some biologically set, some culturally acquired, and that the nonconscious control of action is also an issue to contend with.”

According to the GTI, [3, 4], consciousness is a behavior exhibited by biological systems using their mental structures, and the mental structures model the “self” and its interactions with the material world using the cognitive apparatuses and the information processing structures that transform the information received into operational knowledge using the life processes that are inherited and learned.

The Burgin Mikkilineni (BM) thesis [5] states the ontological BM thesis, which states that the autopoietic and cognitive behavior of artificial systems must function on three levels of information processing systems and be based on triadic automata. The axiological BM thesis states that efficient autopoietic and cognitive behavior has to employ structural machines. The structural machines operate on knowledge structures containing the operational processes which are modeled by named sets.

GTI provides the tools to model autopoietic and cognitive behaviors of biological systems and also provides a path to infuse these behaviors into digital automata. 

The extensive writings of Burgin provide a model for the three levels of intelligence (local, clustered, and global) and mean to model and implement various autopoietic and cognitive behaviors that lead to consciousness that uses the states and dynamics of the system captured as knowledge structures.

Hopefully, the authors find this information useful as I did when I discovered and reflected on it.

[ 1] Dehaene, Stanislas. (2014). “ Consciousness and the Brain: Deciphering How the Brain Codes Our Thoughts” Penguin Group, New York.

[ 2] Damasio, Antonio. Self Comes to Mind. Knopf Doubleday Publishing Group. Kindle Edition.

[ 3] Burgin, M. Theory of Information: Fundamentality, Diversity, and Unification; World Scientific: Singapore, 2010.

[ 4] Burgin, M. Theory of Knowledge: Structures and Processes; World Scientific: New York, NY, USA; London, UK; Singapore, 2016.

[ 5] Mikkilineni, R. A New Class of Autopoietic and Cognitive Machines. Information 202213, 24. https://doi.org/10.3390/info13010024

Author Response

Thank you very much for your very positive review, and for your extensive discussion about GTI and other body of work including by Dehaene and Damasio. Your discussion is certainly relevant and interesting.

(Note: all changes in the manuscript are in red).

Addressing rigorously (that is formally), the relationships between our approach and GTI would be quite an endeavor that would deserve a full paper. We feel that this is really beyond the scope of the current manuscript however, and would not want to add an artificial/superficial brief discussion about it in the manuscript, which would not be satisfying. Since you kindly recommended that the manuscript could be published as it is, we think that you should be ok if we don’t revise the manuscript by including such discussion.

However, we now cite some of the references you provided in the introduction of the revised manuscript in relevant places:

[ 1] Dehaene, Stanislas. (2014). “ Consciousness and the Brain: Deciphering How the Brain Codes Our Thoughts” Penguin Group, New York.

[ 2] Damasio, Antonio. Self Comes to Mind. Knopf Doubleday Publishing Group. Kindle Edition.

[ 3] Burgin, M. Theory of Information: Fundamentality, Diversity, and Unification; World Scientific: Singapore, 2010.

[ 4] Burgin, M. Theory of Knowledge: Structures and Processes; World Scientific: New York, NY, USA; London, UK; Singapore, 2016.

Reviewer 2 Report

This is an excellent overview and up-to date paper about a theory of consciousness, the projective consciousness model (PCM) by Rudrauf and collegues which relies on geometry and subjective perspectives/viewpoints within the world. The theory is introduced, discussed and future empirical and conceptual issues are indicated. This is an excellent overview paper which raises only a few remarks.

-          The concepts of first-, second-, and third-person perspective may want to be explained in more detail and put into the context of their concept of ‘subjective perspective’

-          How do the authors’ concept of geometry stand in relation to the recent paper by Pang et al. (2023) (Nature) on the geometric organisation of the brain?

-          I was wondering that the concept of temporo-spatial alignment as defined as one mechanism of the Temporo-spatial theory of consciousness (TTC) was not considered at all. For my taste, may be I am wrong as I am also not an expert in the TTC, both the PCM and TTC can easily be converged, the most and best point of convergence being in temporo-spatial alignment. The authors of the present paper may want to put their PCM in the context of the TTC and specifically temporo-spatial alignment (and nestedness).

-           Their notion of subjective perspective is very close to the concept of the point of view. I saw one recent paper by Northoff & Smith 2022 (Theory & Psychology) which puts this notion into the context of neuroscience. My feeling is that the present paper could benefit from converging their idea of the subjective perspective and its geometric features with the concept of the point of view as espoused in the Northoff & Smith paper.

-          Some recent review papers about consciousness like Seth & Bayne 2022, Northoff & Lamme 2020, Mashour et al. 2021 may want to be cited

-           

-           

Author Response

Thank you very much for your quite positive review, and for your questions and suggestions.

(Note: all changes in the manuscript are in red).

1) The concepts of first-, second-, and third-person perspective may want to be explained in more detail and put into the context of their concept of ‘subjective perspective’

This is indeed an important as the use of the terms 1PP and 3PP (not mentioning 2PP) have different meaning in specialized literature, e.g. spatial cognition or vision versus consciousness studies (neurophenomenology notably), which we crosspollinate here. Without specification this could be a source of confusion in reading our article, even though we only use 1PP and 3PP in few instances in the manuscript. We added a paragraph in section 1.2 The subjective perspective of consciousness, about this. 

2) How do the authors’ concept of geometry stand in relation to the recent paper by Pang et al. (2023) (Nature) on the geometric organisation of the brain?

The work of Pang et al. (2023) is trying to relate the anatomical geometrical organization of the brain with constraints on the propagation and interaction of electrical fields in the brain. This is very interesting of course, and promising. The geometry and functional processes we invoke cannot be (at least given our current knowledge and understanding) directly related to such principles in any rigorous or meaningful way (by that we mean through a rigorous and unified mathematical formulation). But this is true more generally about any neural model at this point. Our view (at least that of D. Rudrauf) is that the process we are considering involve a form of “virtualization” that we expect to be quite indirectly related to the process we can currently observe and model in the brain (e.g. Rudrauf, 2014). If Neural Field Theory (NFT) is indeed a valid hypothesis, there should be at some point some connection that could be made, but at this point, we have no idea how that would be the case.

We added a brief discussion of this point in the section 4.4 on the Neural Correlates of Consciousness of the discussion.

3) I was wondering that the concept of temporo-spatial alignment as defined as one mechanism of the Temporo-spatial theory of consciousness (TTC) was not considered at all. For my taste, may be I am wrong as I am also not an expert in the TTC, both the PCM and TTC can easily be converged, the most and best point of convergence being in temporo-spatial alignment. The authors of the present paper may want to put their PCM in the context of the TTC and specifically temporo-spatial alignment (and nestedness).

The Temporo-spatial Theory of Consciousness (TTC) is certainly an interesting framework and hypothesis (which would also be interesting to relate somehow to NFT as a possible set of contributing mechanisms). As for your point (2) (see above), the TTC strongly focuses on neural activity at a conceptual and observational level that is still quite remote from the level of modeling we situate (at this point) our approach.

So putting the PCM in the context of TTC would not be well motivated and sound at this point.

Like for point (2), we added a brief discussion of this point in the section 4.4 on the Neural Correlates of Consciousness of the discussion.

4) Their notion of subjective perspective is very close to the concept of the point of view. I saw one recent paper by Northoff & Smith 2022 (Theory & Psychology) which puts this notion into the context of neuroscience. My feeling is that the present paper could benefit from converging their idea of the subjective perspective and its geometric features with the concept of the point of view as espoused in the Northoff & Smith paper.

We have no basis or principled manner at this point to relate the geometry of point of view and perspective we built upon, and more generally the processes we invoke, with the hypothesis that scale-free activity in the brain may underpin such subjective point of view. The hypothesis that such relation exist is certainly interesting, and could be explored in the future, assuming the mathematics connecting all this levels of integration and processing could be rigorously formulated, or perhaps with some clever empirical design, but still after some serious mathematical developments. 

Like for point (2 and 3), we added a brief discussion of this point in the section 4.4 on the Neural Correlates of Consciousness of the discussion.

5) Some recent review papers about consciousness like Seth & Bayne 2022, Northoff & Lamme 2020, Mashour et al. 2021 may want to be cited

Thank you very much for these relevant references, which we added to the introduction.

Round 2

Reviewer 2 Report

all my comments were addressed